# Anti-Inflammatory Effects of Cannabigerol in Rheumatoid Arthritis Synovial Fibroblasts and Peripheral Blood Mononuclear Cell Cultures Are Partly Mediated by TRPA1

**DOI:** 10.3390/ijms24010855

**Published:** 2023-01-03

**Authors:** Torsten Lowin, Marianne Sofia Tigges-Perez, Eva Constant, Georg Pongratz

**Affiliations:** 1Clinic for Rheumatology & Hiller Research Center Rheumatology, University Hospital Duesseldorf, Medical Faculty of Heinrich Heine University, 40225 Duesseldorf, Germany; 2Center for Rheumatologic Rehabilitation, Asklepios Clinic, 93077 Bad Abbach, Germany; 3Medical Faculty, University of Regensburg, 93053 Regensburg, Germany

**Keywords:** rheumatoid arthritis, synovial fibroblasts, cannabinoid, cannabigerol, TRPA1, TNF, IL-6, IL-8, PBMCs, calcium

## Abstract

Since its medical legalization, cannabis preparations containing the major phytocannabinoids (cannabidiol (CBD) and δ^9^-tetrahydrocannabinol (THC)) have been used by patients with rheumatoid arthritis (RA) to alleviate pain and inflammation. However, minor cannabinoids such as cannabigerol (CBG) also demonstrated anti-inflammatory properties, but due to the lack of studies, they are not widely used. CBG binds several cellular target proteins such as cannabinoid and α2-adrenergic receptors, but it also ligates several members of the transient potential receptor (TRP) family with TRPA1 being the main target. TRPA1 is not only involved in nnociception, but it also protects cells from apoptosis under oxidative stress conditions. Therefore, modulation of TRPA1 signaling by CBG might be used to modulate disease activity in RA as this autoimmune disease is accompanied by oxidative stress and subsequent activation of pro-inflammatory pathways. Rheumatoid synovial fibroblasts (RASF) were stimulated or not with tumor necrosis factor (TNF) for 72 h to induce TRPA1 protein. CBG increased intracellular calcium levels in TNF-stimulated RASF but not unstimulated RASF in a TRPA1-dependent manner. In addition, PoPo3 uptake, a surrogate marker for drug uptake, was enhanced by CBG. RASF cell viability, IL-6 and IL-8 production were decreased by CBG. In peripheral blood mononuclear cell cultures (PBMC) alone or together with RASF, CBG-modulated interleukin (IL)-6, IL-10, TNF and immunoglobulin M and G production which was dependent on activation stimulus (T cell-dependent or independent). However, effects on PBMCs were only partially mediated by TRPA1 as the antagonist A967079 did inhibit some but not all effects of CBG on cytokine production. In contrast, TRPA1 antagonism even enhanced the inhibitory effects of CBG on immunoglobulin production. CBG showed broad anti-inflammatory effects in isolated RASF, PBMC and PBMC/RASF co-cultures. As CBG is non-psychotropic, it might be used as add-on therapy in RA to reduce IL-6 and autoantibody levels.

## 1. Introduction

The use of cannabis is on the rise since its medical legalization in many countries including Germany [1]. The most beneficial effects of cannabis extracts are attributed to the action of two major cannabinoids, cannabidiol (CBD) and δ^9^-tetrahydrocannabinol (THC) [2]. However, other non-psychotropic cannabinoids such as cannabigerol (CBG) are still under-researched despite their known efficacy in a variety of conditions [3]. Due to its anti-inflammatory properties, CBG might be suited to treat chronic inflammatory diseases such as rheumatoid arthritis (RA) [4]. RA is a chronic autoimmune disorder that affects around 1% of the general population [5]. It is characterized by autoantibody and pro-inflammatory cytokine production, which eventually leads to the activation of resident synovial fibroblasts (SF) [6]. Rheumatoid arthritis synovial fibroblasts (RASF) produce large amounts of interleukin (IL)-6 but they also engage in matrix degradation by the synthesis of several matrix metalloproteinases (MMPs) such as MMP3 [6]. RASF are activated by tumor necrosis factor (TNF), a major cytokine involved in the pathogenesis of RA. TNF not only induces a general pro-inflammatory phenotype of RASFs but it also up-regulates the expression of transient receptor potential (TRP) ankyrin (TRPA1) [7,8]. TRPA1 was originally described as a nociceptor on sensory neurons [9], but since then, TRPA1 expression was identified in many different tissue and cell types including RASF [8,10]. The role of TRPA1 in non-neuronal cells is still not clarified, but results from tumor cells suggest that TRPA1 activation is a protective mechanism to counteract oxidative stress [11]. In TNF-stimulated RASF, TRPA1 increased intracellular calcium levels and induced cell death upon overactivation with high concentrations of agonists [7,8,12]. Its intracellular localization and calcium mobilizing ability suggest that TRPA1 also influences respiration, autophagy and oxidative stress in RASF [7,8].

In this study, we evaluated the influence of the phytocannabinoid CBG on RASF and lymphocyte function. CBG binds to several target proteins including α_2_ adrenergic receptors, serotonin 5-HT_1A_ receptor, peroxisome proliferator-activated receptor γ, cannabinoid receptor 2 and TRP channels [13]. Within the family of TRP channels, CBG exerts the highest efficacy and potency at TRPA1 [14,15] and, therefore, we investigated the involvement of this ion channel in detail.

## 2. Results

### 2.1. CBG Increases Intracellular Calcium Levels and Compound Uptake

In untreated RASF, CBG (12.5–50 µM) modestly increased intracellular calcium levels (Figure 1a) and this response was not inhibited by the specific TRPA1 antagonist A967079 (Figure 1b) but by the pan TRP inhibitor ruthenium red (RR) (Figure 1c). TRPA1 inhibition even fostered CBG-induced calcium levels (Figure 1b). When RASF were pre-incubated with TNF, CBG (3.125–50 µM) greatly increased intracellular calcium levels (Figure 1d) which was inhibited by the TRPA1 antagonist A967079 (10µM) (Figure 1e) and slightly attenuated by RR (10 µM) (Figure 1f). Of note, A967079 increased basal calcium levels after the 30 min pre-incubation period, whereas RR decreased it (Figure 2k). TNF pre-incubation not only enhanced the potency but also the efficacy of CBG. At the highest concentration of CBG (50 µM), unstimulated RASF increased calcium to 140% (±61.61%) compared to control (Figure 1a), while in TNF-stimulated RASF, calcium was increased to a maximum of 254% (±86.28%) (Figure 1d). In addition, potency was increased as the EC_50_ (at endpoint) shifted from 45.13 µM (unstimulated) to 9.72 µM (TNF stimulated). When extracellular calcium was omitted by using PBS instead of HBSS, CBG alone or together with the TRPA1 inhibitor A967079 did not change calcium levels in unstimulated RASF (Appendix A). RR however, decreased intracellular calcium levels (Appendix A) independent of CBG. In TNF pre-stimulated RASF, CBG enhanced intracellular calcium levels but to a much lower extent compared to HBSS (118% ± 47.48%, PBS vs. 254% ± 86.28%, HBSS at 50 µM CBG) (Appendix A) and this was inhibited by the TRPA1 antagonist at concentrations 3.125 µM, 6.25 µM and 25 µM (Appendix A). In addition to intracellular calcium, we also determined the uptake of the cationic dye PoPo3 iodide, which we recently identified as a surrogate marker for compound uptake coupled to intracellular calcium concentrations [7]. Without TNF pre-stimulation, PoPo3 uptake induced by CBG was negligible (Figure 1g), and, although being significantly influenced by TRPA1 and pan TRP inhibition (Figure 1h,i) effects were small. However, with TNF stimulation, CBG (6.25–50 µM) increased PoPo3 uptake significantly (180% ± 93% at 25 µM) (Figure 1j) which was completely inhibited by the TRPA1 antagonist A967079 (Figure 1k) and modulated but not inhibited by RR (Figure 1l). Without extracellular calcium, PoPo3 uptake was variable and was decreased overall by CBG (Appendix A), whereas the addition of the TRPA1 antagonist A967079 slightly enhanced uptake (Appendix A). TNF stimulation increased CBG-induced PoPo3 uptake (3.125–50 µM) (Appendix A) and this was inhibited by A967079 except for the highest concentration of CBG (50 µM), where uptake was actually increased (Appendix A).

### 2.2. Modulation of CBG-Induced Calcium Levels and PoPo3 Uptake in TNF Stimulated RASF

In the next step, we evaluated whether CBG-induced calcium and PoPo3 uptake can be modulated by short-term (30 min) pre-incubation with either a low concentration of CBG (100 nM), glycyl-l-phenylalanine 2-naphthylamide (GPN, 250 µM), decynium 22 (D22, 10 µM) or ionomycin (Iono, 1 µM). We found that pre-incubation with a low (100 nM) concentration of CBG was able to reduce the effects of subsequent CBG stimulation (Figure 2b) (193% ± 83.3% vs. 164% ± 23.6% (CBG, 100 nM) at 25 µM), suggesting a desensitizing effect on TRPA1. D22 is an inhibitor of organic cation transporters and while it did not modulate basal calcium levels upon pre-stimulation (Figure 2k) it decreased the enhanced intracellular calcium levels induced by CBG (Figure 2c) (193% ± 83.3% vs. 122% ± 22.2% (D22) at 25 µM). GPN and ionomycin at low concentrations (≤1 µM) increase intracellular calcium levels by emptying stores in the endoplasmic reticulum [16,17], and this resulted in higher calcium levels after 30 min of incubation (Figure 2k). Both compounds attenuated the CBG-induced increase in intracellular calcium (Figure 1d,e). Similar results were obtained with PoPo3 as pre-incubation with CBG (100 nM) slightly increased basal uptake of PoPo3 (Figure 2l) but inhibited uptake after subsequent CBG addition (Figure 2g). D22 inhibited CBG-induced PoPo3 uptake (Figure 2h) although pre-incubation initially increased intracellular PoPo3 levels (Figure 2l). GPN abrogated CBG-induced PoPo3 uptake (Figure 2i), but ionomycin enhanced it significantly at 6.25 and 12.5 µM CBG (Figure 2j). In addition, ionomycin also increased basal PoPo3 levels before CBG treatment (Figure 2l).

### 2.3. CBG Reduces Cell Viability, IL-6 and IL-8 Production Dependent on FBS Content

Since CBG had a great impact on calcium levels, we next investigated whether this translates into a modulation of cell viability and cytokine production by RASF. With and without TNF pre-stimulation for 72 h, CBG reduced cell viability of RASF after 24 h incubation, dependent on the FBS content of the medium. Without FBS, CBG, in all concentrations, completely inhibited cytokine production, but this was due to reduced cell viability (Figure 3c,d). However, if the FBS concentration was raised to 2%, CBG inhibited cell viability in higher concentrations (12.5–50 µM) and at 10% FBS only at the highest concentration (50 µM) (Figure 3a,b). IL-6 and IL-8 production were negatively influenced by CBG and this was also dependent on FBS content. Without FBS, TNF-induced IL-6 and IL-8 production was almost completely abrogated by CBG (3.125–50 µM), but the concentration-dependent effects were not observed (Figure 3c,d). At 2% FBS, CBG was less potent and inhibited IL-6 and IL-8 production only in concentrations above 12.5 µM (Figure 3c,d). At 10% FBS, CBG (50 µM) inhibited IL-8 only likely via reduction of cell viability (Figure 3d) but demonstrated a concentration-dependent decrease of IL-6 production (Figure 3c) which was independent of cell viability.

### 2.4. CBG Modulates Cytokine and Immunoglobulin Production in PBMC and RASF/PBMC Co-Cultures

Since CBG blunted IL-6 production by RASF, we also investigated its impact on healthy human peripheral blood mononuclear cells (PBMC) alone and in co-culture with RASF. We stimulated PBMCs with CpG (activates B cells and plasmacytoid dendritic cells), anti-IgM (naïve B cell activator, which is T cell-dependent), anti-CD3/CD28 (T cell activation), anti-CD3/CD28/IgM (naïve B cell and T cell activation) or interferon-γ (IFN-γ, induces human leucocyte antigen (HLA) expression by RASF and subsequent T cell activation due to HLA mismatch). First, we evaluated whether these stimulations have an effect on measured parameters without the addition of CBG. Compared to unstimulated PBMC, we found anti-CD3/CD28, anti-CD3/CD28/IgM and anti-IgM to increase IL-6 production by PBMCs alone (Figure 4a). In PBMC/RASF co-culture, IL-6 levels were more than 100× fold higher due to the presence of RASF and levels were further increased by anti-IgM and decreased by IFN-γ (Figure 4b). IL-10 production was augmented in mono- and co-culture by anti-CD3/CD28, anti-CD3/CD28/IgM, CpG and anti-IgM (Figure 4c,d). TNF was merely absent in unstimulated PBMC but levels were increased by anti-CD3/CD28, anti-IgM and anti-CD3/CD28/IgM in monoculture (Figure 4e) and by anti-CD3/CD28 and anti-CD3/CD28/IgM in co-culture (Figure 4f). Immunoglobulin M (IgM) production was similarly regulated in mono- and co-culture. Whereas anti-CD3/CD28, anti-CD3/CD28/IgM and CpG fostered IgM levels, IFN-γ decreased it (Figure 4g,h). Immunoglobulin G (IgG) was also increased by anti-CD3/CD28, anti-CD3/CD28/IgM and CpG in PBMCs alone, but in co-culture, only CpG enhanced IgG levels while IFN-γ and anti-IgM decreased it (Figure 4i,j). CBG influenced cytokine and immunoglobulin production depending on the stimulation. CBG at 2.5 µM had only a slight stimulatory effect on anti-IgM-induced IL-10 production by PBMCs alone (Figure 4c), but it decreased IgG production in co-culture under control conditions and anti-CD3/CD28, anti-CD3/CD28/IgM and CpG stimulation (Figure 4j). However, CBG at 25µM had a much greater impact on cytokine and immunoglobulin production. Anti-CD3/CD28/IgM-induced IL-6 production by PBMCs was augmented, while basal IL-6 production, CpG and IFN-γ-induced IL-6 were reduced by CBG (Figure 4a). In co-cultures, only anti-IgM-induced IL-6 was reduced (Figure 4b). CpG-induced IL-10 production was decreased in mono- and co-culture whereas anti-IgM-induced IL-10 was increased (Figure 4c,d) and anti-CD3/CD28/IgM-induced IL-10 was slightly reduced in monoculture only (Figure 4c). TNF production was augmented in both culture conditions by anti-CD3/CD28/IgM (Figure 4e,f) and by anti-CD3/CD28 in co-culture (Figure 4e). IgM levels in monoculture were reduced by CBG under all conditions whereas in co-culture, IgM was decreased only in anti-CD3/CD28, anti-CD3/CD28/IgM, CpG stimulated and untreated cells (Figure 4g,h). Similarly, CBG reduced IgG production under all conditions (Figure 4i) except IFN-γ in monoculture and IFN-γ and anti-IgM in co-culture (Figure 4j).

### 2.5. Is TRPA1 Involved in the Regulation of Cytokine and Immunoglobulin Production by CBG?

As CBG modulated cytokine, IgM and IgG production, we assessed the involvement of TRPA1 by using the antagonist A967079 to inhibit the effects of CBG. We analyzed those groups where CBG had an influence on the production of cytokines or immunoglobulins (Figure 4). Of note, A967079 had no own effects under control conditions without CBG (Appendix A) except for the reduction in IgG production in co-culture (Appendix A).

In PBMC monoculture, CBG and anti-IgM-induced IL-6 production was not reduced but rather increased by TRPA1 inhibition, while basal IL-6 production was reduced by CBG and rescued by TRPA1 inhibition (Figure 5a). In co-culture under CpG stimulation, CBG attenuated IL-6 production, and this was reversed by TRPA1 inhibition (Figure 5f). IL-10 production was reduced by CBG in anti-CD3/CD28/IgM-stimulated PBMCs and TRPA1 inhibition further reduced its levels (Figure 5b). However, CBG fostered IL-10 in anti-IgM-stimulated PBMCs, and this was reversed by TRPA1 inhibition (Figure 5b). In co-culture, CpG-induced IL-10 production was attenuated by CBG and further reduced by TRPA1 inhibition (Figure 5f). In addition, increased TNF production by CBG in anti-CD3/CD28-stimulated co-culture was inhibited by A967079 (Figure 5h). CBG reduced IgG production, but this was unaltered by TRPA1 inhibition under all conditions in mono- and co-culture (Appendix A). IgM production was also attenuated by CBG and further reduced by TRPA1 inhibition in monoculture under all conditions except CpG (Figure 5c). In co-culture, CBG attenuated anti-CD3/CD28-induced and basal IgM production and this was further reduced by TRPA1 inhibition (Figure 5g).

## 3. Discussion

In this study, we demonstrated that CBG enhances intracellular calcium levels and PoPo3 uptake by RASF in a TRPA1-dependent manner. In addition, we showed that CBG reduced cell viability dependent on FBS content in the culture medium, and it attenuated TNF-induced IL-6 and IL-8 production by RASF. In PBMCs and PBMC/RASF co-culture, we found CBG to modulate IL-6, IL-10, TNF, IgM and IgG production dependent on activation stimulus.

### 3.1. CBG Modulates Intracellular Calcium and PoPo3 Uptake via TRPA1 (Figure 1 and Figure 2)

In previous studies, we found that TNF up-regulates TRPA1 protein in RASF which is associated with enhanced intracellular calcium levels and PoPo3 uptake in response to TRPA1 activation [8,12]. In line with these results, we detected only a slight influence of CBG on unstimulated RASF in TNF pre-stimulated RASF, and these effects of CBG on calcium and PoPo3 uptake were antagonized by TRPA1 inhibition. CBG is an agonist on several TRP channels, but it has the highest affinity and efficacy at TRPA1 [14,15,18]. Since we detected a CBG-induced increase in intracellular calcium without TNF stimulation, it is likely that CBG also engages other TRP channels (e.g., TRPV1-4). While the TRPA1 antagonist A967079 did not inhibit the effects of CBG in unstimulated RASF, it was efficacious under TNF-pre-stimulated conditions. On the opposite, RR was able to reduce calcium levels in unstimulated RASF, but it was far less potent in TNF pre-stimulated RASF. Although RR is considered a pan TRP inhibitor, it also interferes with the mitochondrial and endoplasmic reticulum (ER) calcium transport at the concentration used in this study but due to its negative charge, RR does not readily penetrate cells and requires permeabilization [19,20]. Taken together, RR inhibits TRP channels at the plasma membrane, while TRPA1 is likely located intracellularly and is only inhibited by the lipophilic antagonist A967079.

We also pre-incubated RASF with a low concentration of CBG which reduced the subsequent increase of calcium and PoPo3 after the addition of higher concentrations of CBG. This suggests that CBG is able to desensitize TRPA1, although at lower concentrations than reported in the literature [15]. However, it might be possible that intracellular concentrations of CBG are much higher than 100 nM due to accumulation in membranes or binding to fatty acid binding proteins as described for cannabidiol and tetrahydrocannabinol [21]. As we found CBG to increase calcium in calcium-free phosphate-buffered saline, which suggests mobilization from intracellular stores, we also depleted ER calcium stores by using glycyl-l-phenylalanine 2-naphthylamide (GPN) or ionomycin [16]. We used a concentration of ionomycin that is reported to only permeabilize the ER membrane [17], but since this study was performed on human umbilical vein endothelial cells, it could be that in RASF ionomycin also elicits calcium influx via the plasma membrane. GPN, however, increased basal calcium levels, likely via depletion of ER calcium stores, and this attenuated the effects of CBG on calcium and precluded the uptake of PoPo3. This uptake is mediated via organic cation transporters (OCT) [7] as decynium-22, an inhibitor of several OCT isoforms and related transporters abrogated PoPo3 accumulation. GPN also blunted PoPo3 uptake, possibly via its ability to elevate intracellular pH [16] as OCT-mediated uptake is pH-dependent [22].

### 3.2. CBG Reduces Cell Viability and Cytokine Production in an FBS-Dependent Fashion (Figure 3)

Elevation of intracellular calcium levels in RASF might be anti-inflammatory [23] and, therefore, we also assessed cell viability, IL-6 and IL-8 production by RASF. Similar to our previous findings with CBD [7], CBG reduced cell viability in an FBS-dependent fashion. High FBS levels protected RASF from CBG-induced cell death, likely by scavenging free CBG, as cannabinoids in humans also bind to serum albumin [24]. Likewise, IL-6 production was inhibited by CBG, and this correlated well with reduced cell viability under low or no FBS conditions. However, at 10% FBS, CBG showed a concentration-dependent inhibitory effect on IL-6 production, which was independent of the reduction in cell viability. CBG and its derivatives show an anti-inflammatory effect in several disease models [25,26,27], but besides a partial involvement of the cannabinoid receptor 2 (CB_2_) [27], it is unclear by which mechanism CBG elicits beneficial effects. However, own unpublished data show no influence of CB_2_ on cytokine production, and Fechtner et al. even demonstrate the pro-inflammatory effects of CB_2_ activation in RASF [28].

### 3.3. Influence of CBG on PBMC and PBMC/RASF Co-Culture (Figure 4a–f)

In PBMCs and PBMC/RASF co-culture, we found CBG to modulate cytokine and immunoglobulin production which was dependent on activation stimulus. Of note, without CBG treatment, cytokine and immunoglobulin levels were differentially regulated by activation. IL-6 levels were increased in PBMC alone when T cells or plasmacytoid dendritic cells (pDCs) along with B cells were activated and this was also confirmed by others [29,30]. CBG further increased IL-6 levels induced by anti-CD3/CD28, but reduced IL-6 when unstimulated or stimulated with CpG, IFN-γ or anti-IgM. These effects were blunted in co-culture as RASF are the main producers of IL-6 and might shield the effects of PBMC-derived IL-6. IFN-γ reduced IL-6 production in co-culture, and this was likely due to the inhibitory effect of IFN-γ on RASF which has been described previously [31]. In mono- and co-culture, the anti-inflammatory IL-10 was increased by T cell activation, CpG and anti-IgM, and while CpG-induced IL-10 was reduced by CBG, anti-IgM-induced IL-10 was elevated. CpG induces IL-10 production in B cells but not pDCs [32,33], suggesting an inhibitory effect of CBG on B cells stimulated T-independently. TNF was barely detectable but was strongly induced by T cell activation as previously shown [34] and CBG further increased its production. B cell-specific stimuli did not elicit TNF production, although B cells are capable of producing this cytokine [35,36,37]. However, B cells make up only 5–10% of all PBMCs [38] and this cell number might be too little to detect TNF production. Immunoglobulin production was augmented by T cell activation or CpG in monoculture, and CpG in co-culture and CBG had an inhibitory influence on antibody production under all stimulatory conditions except for IFN-γ. This further strengthens the notion that CBG negatively regulates B cell function.

### 3.4. Does CBG Act on TRPA1 in PBMCs? (Figure 5a,b,d,e,f,h)

We evaluated the effects of CBG on PBMCs and PBMC/RASF co-culture under TRPA1 inhibition. Depending on stimulation, CBG increased or decreased IL-6, IL-10 and TNF production and these effects were either inhibited or augmented by TRPA1 inhibition. This suggests that CBG via TRPA1 does not influence lymphocytes in general but targets only distinct (sub)populations. For instance, anti-CD3/CD28/IgM activates T cells and B cells and under these conditions, CBG reduced IL-10 production which was further decreased by TRPA1 inhibition. On the opposite, anti-IgM-induced (targeting B cells exclusively) IL-10 production was further enhanced by CBG and this was inhibited by TRPA1 antagonism. In previous studies, we already investigated the impact of CBD on CpG-stimulated B cells and found that the TRPA1 antagonist did not reverse the effects of CBD but rather supported them [39]. TRPA1, CBG is also a potent alpha 2 adrenergic and CB_2_ agonist and a moderately efficacious serotonin HT_1a_ receptor antagonist [40]. Therefore, effects that were insensitive to TRPA1 inhibition might be mediated by other target receptors of CBG. In fact, it has been shown that CB_2_ engagement on leucocytes increases IL-6 and IL-10 production [41], HT_1a_ receptors control lymphocyte proliferation [42] and alpha 2 adrenergic receptor agonists increase splenic IL-6 and IFN-γ levels [43]. In addition, it might be that, similar to the action of CBD and other phytocannabinoids, CBG induces mitochondrial dysfunction and apoptosis/necrosis in leucocyte subsets [7,44].

### 3.5. TRPA1 Influences Antibody Production (Figure 4g–j, Figure 5c,g and Appendix A)

IgM and IgG production was reduced by CBG and further decreased by TRPA1 antagonism. However, until now no functional studies involving TRPA1 were conducted in B cells but results from Soutar et al. suggest that TRPA1 is involved in the reduction of IL-6, IL-10 and immunoglobulin production induced by piperine, a TRPV1/TRPA1 agonist [45]. The unexpected result that CBG and A967079 have additional effects might be due to the similar long-term effects of TRPA1 agonists and antagonists as TRPA1 desensitizes upon agonist treatment [46] and this might be required for the effects on immunoglobulin production. However, desensitization still allows for limited TRPA1 activation and complete inhibition might only be achieved by “real” antagonism rather than functional antagonism by desensitization.

## 4. Materials and Methods

### 4.1. Patients

In this study, 33 patients with long-standing RA fulfilling the American College of Rheumatology revised criteria for RA [47], who underwent elective knee joint replacement surgery, were included. Mean age was 70 ± 8 years for RA. Mean C-reactive protein (CRP) was 43 ± 145 mg/L for RA. Rheumatoid factor was 141 ± 243 IU/mL in RA. In the RA patient group, 8/33 received methotrexate, 12/33 glucocorticoids, and 4/33 received biologicals or Janus kinase inhibitors. All patients in this study were informed about the purpose and gave written consent before surgery. This study was approved by the Ethics Committees of the University of Düsseldorf (approval number 2018-87-KFogU and 2018-296-KFogU).

### 4.2. Compounds and Antibodies

Compounds and antibodies with abbreviation, order number, company and concentration used are presented in Table 1.

### 4.3. Synovial Tissue Preparation and SFs Culture

The RASF isolation and preparation were performed as described previously for in-vitro experiments [48]. In brief, synovial tissue samples were immediately collected (up to 9 cm^2^) upon exposing knee joint capsule. Tissue pieces were carefully shredded into tiny fragments and digested with liberase (Roche Diagnostics, Mannheim, Germany) overnight at 37 °C. The filtration (70 µm) and centrifugation (300× *g*, 10 min) of the resulting suspension were carried out subsequently. After that, the pellet was obtained, which was then treated with erythrolysis buffer (20.7 g NH_4_Cl, 1.97 g NH_4_HCO_3_, 0.09 g EDTA and 1 L H_2_O) for 5 min. The suspension was centrifuged again for 10 min at 300× *g*. At last, pelleted RASFs were resuspended in phenol red-free RPMI-1640 (Sigma Aldrich, Taufkirchen, Germany, #R7509) with 10% FBS (Gibco/Thermo Fisher, Schwerte, Germany #10500-064), 1% penicillin/streptomycin (Sigma, #P4333-100ML), 1% GlutaMax (Thermo Fisher, #35050-038), 1% Sodium pyruvate (Sigma, #S8636) and 25 mM HEPES (Sigma, #H0887-100 mL). Cells were maintained at 37 °C and 5% CO_2_. After culturing overnight, cells were treated with fresh medium to wash off dead cells and debris.

### 4.4. Isolation of PBMCs from Peripheral Blood

PBMCs were isolated using the Greiner LeucoSep Tubes (#227290, Greiner Bio-one) according to manufacturers’ instructions.

### 4.5. RASF Co-Culture with PBMCs

Co-culture experiments were performed in 96 well plates (Cellstar, Greiner bio-one, Kremsmünster, Austria). In brief, 5000 RASF were seeded in 200 µL RPMI-1640 with 10% FBS (Thermo Fisher/Gibco, #10500-064) and grown for 72 h. Then, growth medium was replaced by fresh RPMI with 10% FBS and 250,000 isolated human PBMCs were added. Cells were stimulated with cytokines/CBG as indicated for 7d in RPMI medium with 10% FBS. After that, supernatants were collected and cytokine and immunoglobulin production was assessed by ELISA.

### 4.6. IL-6, IL-8, IL-10 and TNF ELISA and Stimulation of SF

ELISA kits for IL-6 (#555220), IL-8 (#555244), IL-10 (#555157) and TNF (#555212) were purchased from BD (Franklin Lakes, NJ, USA) and were conducted according to the manufacturers’ instructions. In 96-well plates, 5000 RASF/well were seeded in 200 µL RPMI-1640 with 10% FBS and grown for 72 h. Then, growth medium was replaced by fresh RPMI (2% FBS), and SFs were primed with TNF (10 ng/mL) for 3 days to induce TRPA1 protein. After that, culture medium was replaced with RPMI (2% FBS) and CBG was added for an additional 24 h. After that, supernatants were collected and analyzed. All samples were run in duplicates.

### 4.7. IgM and IgG ELISA

96 well MaxiSorp plates (Thermo Fisher Scientific, Waltham, MA, USA) were coated with 100 µL [10 µg/mL] of affiniPure goat anti-human IgG (H+L) or affiniPure goat anti-human IgM, Fc5μ fragment specific, diluted in PBS. Plates were sealed with adhesive strips and incubated at 4 °C overnight. Then, plates were washed twice using 200 µL per well of wash buffer (PBS containing 0.05% Tween-20) and blocked with 200 µL blocking buffer (1% BSA, 5% sucrose in PBS) for 1 h at RT. Standards were prepared from IgG or IgM from human serum (see Table 1). Standard concentrations were 200 ng/mL diluted in culture medium. A 1:2 serial dilution in assay buffer (Blocking Buffer diluted 1:2 in PBS) was carried out to obtain 7 standard concentrations ranging from 100 ng/mL to 1.5625 ng/mL. Moreover, 50 µL of each standard was added to the respective wells. Notably, 25 µL of assay buffer was added to all sample wells. Moreover, 25 µL of co-culture supernatant was added, and the plate was sealed with adhesive film and incubated at RT for 2 h on a microplate shaker at 400 rpm. After incubation, plates were washed four times as previously described. Then, detection antibodies (peroxidase affiniPure goat Anti-human IgG (H+L) or IgM, Fc5μ fragment specific) were diluted 1:50,000 in assay buffer. Moreover, 50 µL of diluted detection antibody was added per well and incubated for 1 h at RT. After washing four times, 50 µL of 3,3’’,5,5’’-Tetramethylbenzidine (Ultra TMB, Thermo, # 34028) substrate solution was added to each well, and the plate was incubated for 30 min. The reaction was stopped by adding 50 µL 2N sulphuric acid. The plate was read at a wavelength of 450 nm with a reference wavelength of 595 nm in a TECAN Infinite M200 Pro plate reader.

### 4.8. Intracellular Calcium and PoPo3 Uptake

In black 96-well plates, RASF were incubated with 4 µM of calcium dye Cal-520 (ab171868, abcam, Cambridge, UK) in Hanks buffered salt solution (1.66 mM Ca^2+^) (HBSS, sigma, # 55037C) or PBS (no Ca^2+^) with 0.02% Pluoronic F127 (Thermo fisher scientific, Waltham, MA, USA, # P6866) for 60 min at 37 °C followed by 30 min at room temperature. After washing, HBSS or PBS containing 1 µM PoPo3 iodide (Thermo fisher scientific, # P3584) and respective antagonists/ligands/inhibitors were added for 30 min at room temperature. After that, CBG was added and the intracellular Ca^2+^ concentration as well as PoPo3 uptake were evaluated with a TECAN multimode reader over 90 min.

### 4.9. Cell Viability Assay

After collecting the supernatants of treated RASFs, cells were incubated with CellTiter-Blue reagent following the instruction of the manufacturer (G8081, Promega, Madison, WI, USA). By determining the reduction from resazurin to resorufin, cell viability was quantified to reflect the toxic effect of CBG.

### 4.10. Statistics

All data were presented from at least three independent experiments. SPSS 27 (IBM, Armonk, NY, USA) was used for data analysis. The statistical tests used are given in the figure legends. When data are presented as line plots, the line represents the mean. When data are presented as bar charts, the top of the bar represents the mean, and error bars depict the standard error of the mean (SEM). When data are presented as box plots, the boxes represent the 25th to 75th percentiles, the lines within the boxes represent the median, and the lines outside the boxes represent the 10th and 90th percentiles. The level of significance was *p* < 0.05.

## 5. Conclusions

In this study, we evaluated the effect of CBG on isolated RASF and PBMCs alone and in co-culture with RASF. We found robust anti-inflammatory effects on cytokine production, cell viability and antibody production. Since its medical legalization, cannabis research focused on THC and CBD but we provide evidence that CBG might be even superior to the aforementioned compounds as shown previously [24,42]. CBG has some advantages over THC and CBD when used therapeutically: In contrast to THC, CBG is non-psychotropic and shows broader anti-inflammatory effects as THC did not modulate IL-6 production by RASF alone [12]. CBD on the other hand has been shown to eliminate RASF by a calcium overload in vitro [7], drive B cell apoptosis and reduce PBMC cytokine production [34]. These effects were not mediated by specific receptor interactions but rather by modulating mitochondrial ion transport. Therefore, CBG might be suited as an adjunct therapy for RA to reduce cytokine and autoantibody production.

## Figures and Tables

**Figure 1 ijms-24-00855-f001:**
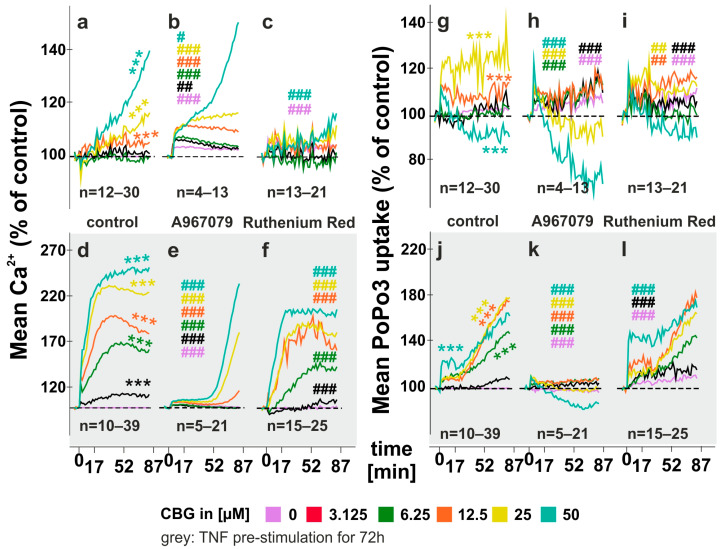
Intracellular calcium levels and PoPo3 uptake of RASF under the influence of CBG Influence of CBG on intracellular calcium levels (**a**–**f**) and PoPo3 uptake (**g**–**l**). RASF was simulated (**d**–**f**;**j**–**l**) or not (**a**–**c**;**g**–**i**) with TNF for 72 h, and assays were conducted with extracellular calcium by using Hanks balanced salt solution (HBSS). The TRPA1 inhibitor A967079 or pan TRP inhibitor ruthenium red were added 30 min prior to the addition of CBG. *** *p* < 0.001 for differences between concentrations of CBG. ANOVA with Dunnett’s T3 post hoc test was used for comparisons. # *p* < 0.05, ## *p* < 0.01, ### *p* < 0.001 for comparisons of CBG versus CBG/antagonist treatment at each concentration. ANOVA with the Bonferroni post hoc test was used for comparisons. Number of patients included: (**a**,**g**) n = 12 and n = 30 (no CBG); (**b**,**h**) n = 4 and n = 13 (no CBG) (**c**,**i**) n = 13 and n = 21 (no CBG) (**d**,**j**) n = 10 (50 µM), n = 12 (3.125–25 µM) and n = 39 (no CBG); (**e**,**k**) n = 5 and n = 21 (no CBG) (**f**,**l**) n = 15 and n = 25 (no CBG).

**Figure 2 ijms-24-00855-f002:**
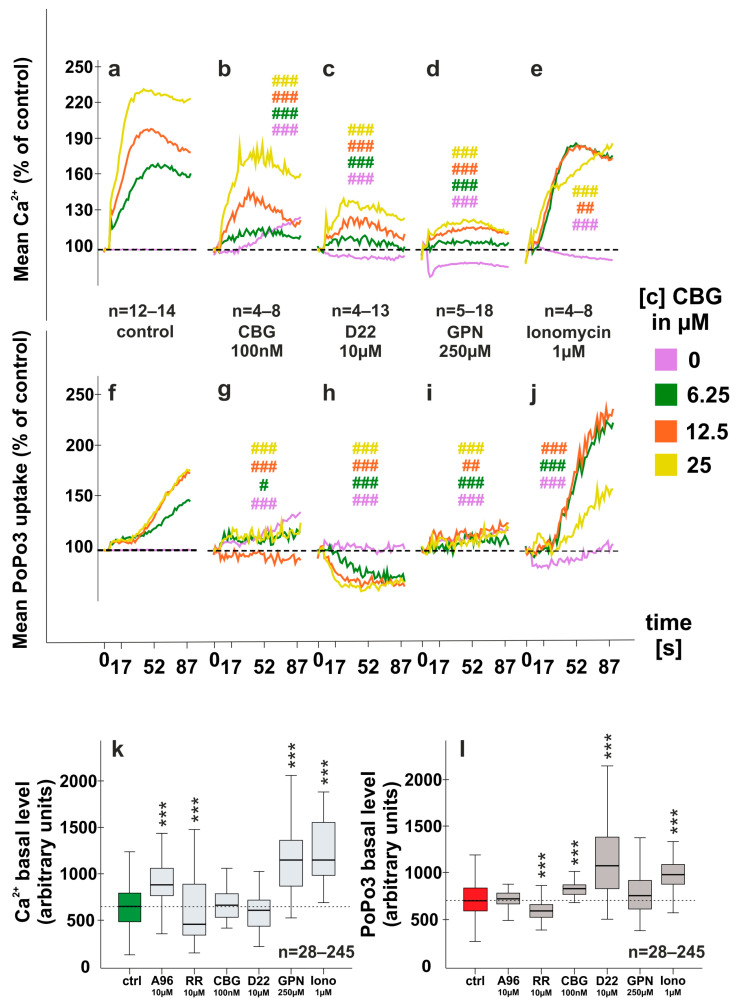
Modulation of CBG-induced calcium and PoPo3 uptake of RASF. (**a**–**j**) CBG-induced calcium (**a**–**e**) and PoPo3 uptake (**f**–**j**) are modulated by 30 min pre-incubation with either CBG (100 nM) (**b**,**g**), the organic cation transport inhibitor decynium-22 (D22) (**c**,**h**), the endoplasmic calcium releaser glycyl-l-phenylalanine 2-naphthylamide (GPN) (**d**,**i**) or the calcium ionophore ionomycin (**e**,**j**). (**k**,**l**) Absolute calcium (**k**) and PoPo3 levels (**l**) after 30 min incubation with the compounds indicated. # *p* < 0.05, ## *p* < 0.01, ### *p* < 0.001 for comparisons of CBG versus CBG/antagonist treatment at each concentration. ANOVA with Bonferroni post hoc test was used for comparisons (**a**–**j**). *** *p* < 0.001, for comparisons between different 30 min pre-incubation conditions before the addition of CBG. Kruskal-Wallis with Bonferroni post hoc test was used for comparisons (k, **l**). RR = ruthenium red; A96 = A967079. Number of patients included: (**a**,**f**) n = 27 and n = 39 (no CBG); (**b**,**g**) n = 4 and n = 8 (no CBG); (**c**,**h**) n = 4 and n = 13 (no CBG) (**d**,**i**) n = 5 and n = 18 (no CBG) (**e**,**j**) n = 4 and n = 8 (no CBG). Number of experimental replicates from patient samples (see number of patients in Figure 1d–f and Figure 2a–j): (**k**,**l**) n = 245 (ctrl), n = 46 (A96), n = 100 (RR), n = 28 (CBG), n = 33 (D22), n = 43 (GPN), n = 28 (Iono).

**Figure 3 ijms-24-00855-f003:**
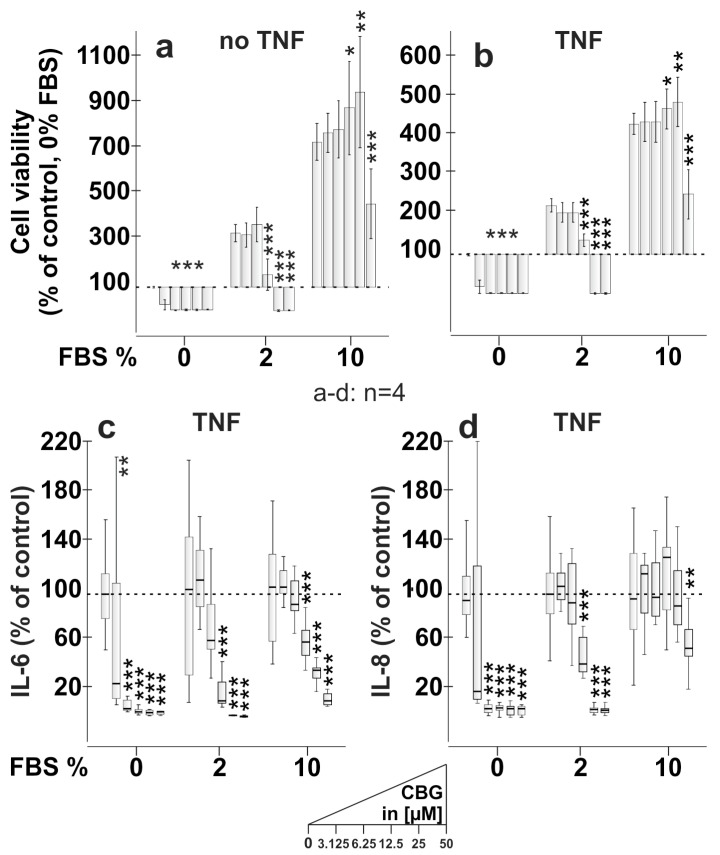
Influence of CBG on cell viability, IL-6 and IL-8 production of RASF**.** (**a**,**b**) Cell viability of RASF after stimulation with CBG for 24 h, either with (**b**) or without (**a**) 72 h pre-treatment with TNF [10 ng/mL] in culture medium containing 0%, 2% or 10% FBS. The dotted line at 100% represents the control. All values were normalized to the control at 0% FBS (**a**,**b**). (**c**,**d**) Reduction of TNF-induced IL-6 (**c**) and IL-8 (**d**) production by CBG in culture medium containing 0%, 2% or 10% FBS. The dotted line represents the control. Control IL-6 levels were 4.1 ng/mL ± 3.1 ng/mL (0% FBS), 23.5 ng/mL ± 24.6 ng/mL (2% FBS) and 18.6 ng/mL ± 12.4 ng/mL (10% FBS). Control IL-8 levels were 14.4 ng/mL ± 5.0 ng/mL (0% FBS), 38.9 ng/mL ± 16.1 ng/mL (2% FBS) and 44.0 ng/mL ± 26.7 ng/mL (10% FBS). *** *p* < 0.001, ** *p* < 0.01, * *p* < 0.05 for differences between concentrations of CBG vs. control. Block design ANOVA with Dunnett’s post hoc test was used for comparisons. The box plots of each series (0, 2 and 10% FBS) represent 0 µM, 3.125 µM, 6.25 µM, 12.5 µM, 25 µM and 50 µM CBG.

**Figure 4 ijms-24-00855-f004:**
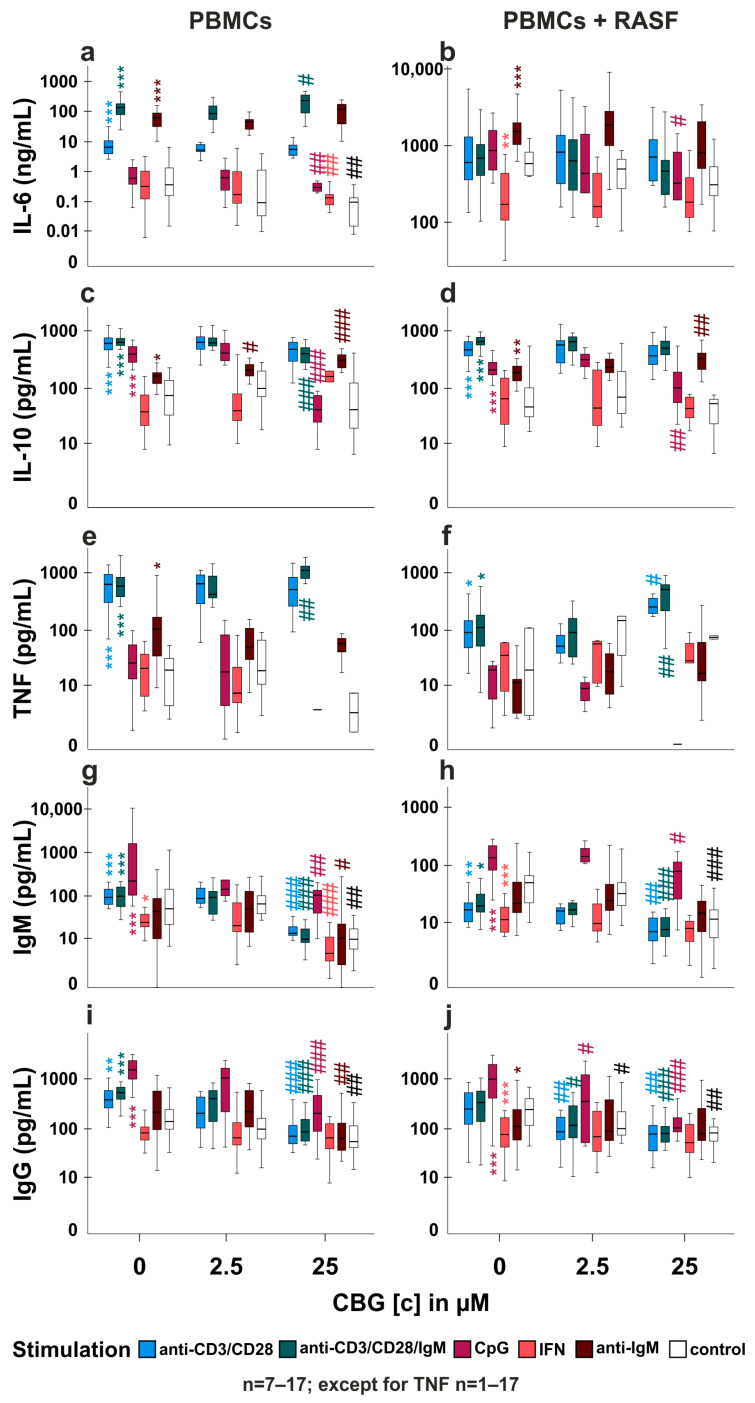
Influence of CBG on IL-6, IL-10, TNF, IgM and IgG production by PBMCs and PBMC/RASF co-cultures. (**a**–**j**) PBMCs alone (**a**,**c**,**g**,**e**,**i**) or PBMC/RASF co-cultures (**b**,**d**,**h**,**j**) were activated with anti CD3/CD28 (T cells), anti-CD3/CD28/IgM (T cells and B cells), CpG (B cells), anti-IgM (B cells) or IFN-γ (T cells) and concomitantly stimulated with CBG [2.5 µM or 25 µM] for 7 days in medium with 10% FBS. Thereafter, IL-6 (**a**,**b**), IL-10 (**c**,**d**), TNF (**e**,**f**), IgM (**g**,**h**) and IgG (**i**,**j**) were determined. Cytokine levels are presented on a logarithmic scale. *** *p* < 0.001, ** *p* < 0.01, * *p* < 0.05 for differences between treatment without CBG. Kruskal-Wallis test with Bonferroni post hoc test was used for comparisons. # *p* < 0.05, ## *p* < 0.01, ### *p* < 0.001 for comparisons of control versus CBG treatment at a given stimulation. Kruskal-Wallis test with Bonferroni post hoc test was used for comparisons.

**Figure 5 ijms-24-00855-f005:**
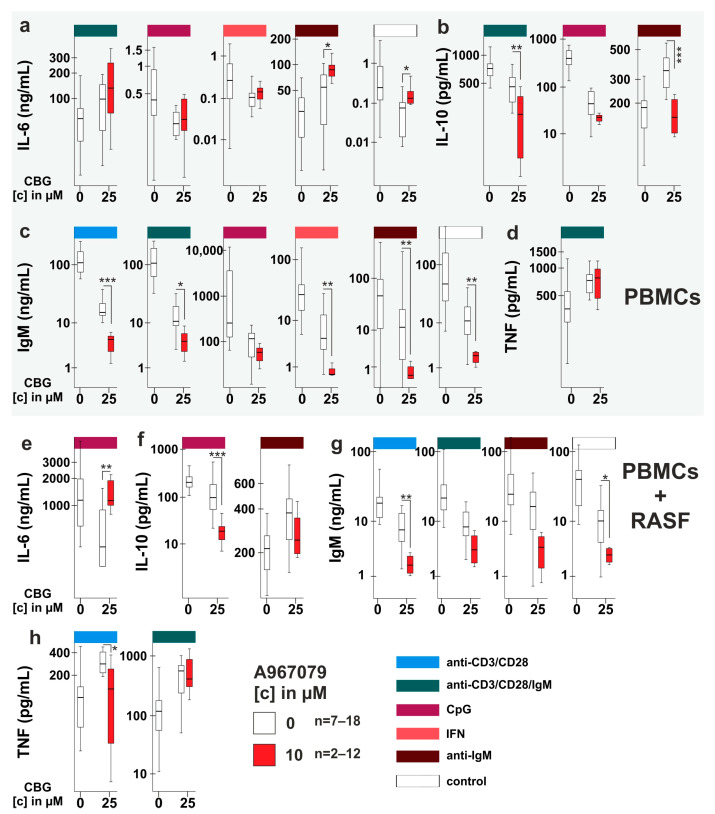
TRPA1 involvement in cytokine and antibody production modulated by CBG in PBMCs and PBMC/RASF co-cultures. (**a**–**h**) PBMCs alone (**a**–**d**) or PBMC/RASF co-cultures (**e**–**h**) were co-incubated with CBG [25 µM] (controls are shown for comparison) and the TRPA1 inhibitor A967079 [10 µM] under different activation stimuli for 7 days. Cytokine levels are presented on a logarithmic scale. *** *p* < 0.001, ** *p* < 0.01, * *p* < 0.05 for differences between treatment with A9607079/CBG versus CBG alone. Mann Whitney U-test was for comparisons. Number of patients included: (**a**) For αCD3/CD28/IgM and αIgM: n = 12 (A96) and n = 18; for CpG: n = 12 (A96) and n = 15; for IFN: n = 10 (A96) and n = 17; for control: n = 11 (A96) and n = 14. (**b**) For αCD3/CD28/IgM and for αIgM: n = 8 (A96) and n = 18; for CpG: n = 3 (A96) and n = 18. (**c**) For αCD3/CD28, αCD3/CD28/IgM and for αIgM: n = 4 (A96) and n = 18; for CpG and IFN: n = 3 (A96) and n = 16; for control: n = 4 (A96) and n = 17. (**d**) n = 4 (A96) and n = 17. (**e**) n = 8 (A96) and n = 18. (**f**) for CpG: n = 7 (A96) and n = 18; for αIgM: n = 8 (A96) and n = 17. (**g**) For αCD3/CD28, αCD3/CD28/IgM, αIgM and IFN: n = 4 (A96) and n = 17; for CpG: n = 2 (A96) and n =15; for control: n = 4 (A96) and n = 18.

**Table 1 ijms-24-00855-t001:** Compounds and antibodies used in this study.

Name of Compound/Antibody	Abbreviation (if Applicable)	Order #	Company
A967079	A96	4716	Bio-Techne/Tocris, Wiesbaden-Nordenstadt, Germany
Ruthenium red	RR	1439	Bio-Techne/Tocris, Wiesbaden-Nordenstadt, Germany
Cannabigerol	CBG		THC pharm, Frankfurt, Germany (discontinued)
Decynium-22	D22	4722	Bio-Techne/Tocris (discontinued)
Ionomycin	Iono	1704	Bio-Techne/Tocris, Wiesbaden-Nordenstadt, Germany
glycyl-l-phenylalanine 2-naphthylamide	GPN	14634	Biomol/Cayman, Hamburg, Germany
anti-CD3		16-0039-81	VWR International/Thermo Fisher/Life Sc., Darmstadt, Germany
anti-CD28		16-0289-81	VWR International/Thermo Fisher/Life Sc., Darmstadt, Germany
CpG Oligonucelotide	CpG	tlrl-2006-1	Invivogen, Toulouse, France
Interferon-gamma	IFN-γ	300-02	Peprotech/Thermo Fisher Scientific, Hamburg, Germany
Tumor necrosis factor	TNF	300-01A	Peprotech/Thermo Fisher Scientific, Hamburg, Germany
Goat Anti-Human IgM		109-006-129	Dianova/Jackson ImmunoResearch, Hamburg, Germany
IgG from human serum		I2511	Sigma-Aldrich, Taufkirchen, Germany
IgM from human serum		I8260	Sigma-Aldrich, Taufkirchen, Germany
Peroxidase AffiniPure Goat Anti-Human IgG (H+L)		109-035-003	Dianova/Jackson ImmunoResearch, Hamburg, Germany
Goat anti-Human IgG, IgM, IgA (H+L) Secondary Antibody		31128	Invitrogen/Thermo Fisher Scientific, Schwerte, Germany

## Data Availability

The datasets used and/or analyzed during the current study are available from the corresponding author on reasonable request.

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
