# Peer review of "Anti-Inflammatory Effects of Cannabigerol in Rheumatoid Arthritis Synovial Fibroblasts and Peripheral Blood Mononuclear Cell Cultures Are Partly Mediated by TRPA1"

_ijms, 2023, doi:10.3390/ijms24010855_

Round 1

Reviewer 1 Report

In this manuscript, Lowin et al. study the effects of the non-psychotropic cannabinoid CBG on rheumatoid synovial fibroblasts and peripheral blood mononuclear cells. They explore the extent to which CBG mediates the anti-inflammatory effects and how it affects cell viability and antibody production. Morover, the authors explore how TRPA1 is involved in these processes.

This work was performed with a solid technical competence and the questions addressed by these experiments are clear and interesting. On the other hand, the work suffers from a cluttered graphical presentation of a large number of interesting results. I would recommend reworking the graphic design of the figures so that the meaning of the results is apparent at a glance and the use of colour is strictly purposeful. In addition, because the study contains a number of results whose significance is not entirely clear from the results section, I recommend revising the discussion.

1)           The exact sample size (n, not just the span) for each experimental group should be given. This applies to all data presented in Figures 1-5. Wherever the mean and standard error are given in the text, n must also be given.

2)           Overall, I especially don't like the graphic design of the Figures:

a)           In my copy of the manuscript, the ticks on both axes are lacking in all figures. The correct axis designation must be completed. In addition, I don't understand why there are arrows on the axes in Figs. 1, a to j, and Fig. 2.

b)           In Figures 1, 2, and 3: The time intervals (1038, 3115) that are not equidistant are very unusually chosen. Maybe it would be better to convert sec to min?

c)            Moreover, in Figs. 1-3, the labeling of the individual panels with small letters in the middle of the graphs is not clear. I recommend to label the individual panels as it is usually done, on the top left.

d)           In the legends to all figures, the concentration is given as [c]. This concentration designation is confusing and is not commonly used. At first glance, the legend appears to refer to panel c. I recommend omitting this symbol in the legends and use „concentration“ instead of „[c]“ in the text to figures.

e)           In the experiments illustrated in Figures 1 and 2, the signals exhibit excessive „noise“ (e.g. panels s,t in Fig. 1). I suspect that the records were exported to some graphics program without rounding the edges of the connecting lines in the software environment, which artificially adds unnecessary variance.

f)            Figure 3, Comparison of the viability of TNF-exposed and non-exposed cells would be more clear if the y-axes scales were the same in panels a and b. „FBS“ in y-axis description in Figure 3a ? I'm wondering if it might be better to show the y-axis logarithmically in Fig. 3c,d? Why are columns a,b not outlined while c,d are? Are

g)           The labeling of the preincubated CBG in Figure 2b and g is confusing. The low concentration should be explicitly indicated (by indicating 100 nM) and the sentence in the legend to Fig. 2 k and l should be worded differently (not „…without the addition of CBG“)

3)           Throughout the text, I recommend writing specifically TNF-alpha instead of TNF or at least introducing the abbreviation in the abstract/introduction.

4)           Line 27, the abbreviation PBMC must be introduced here

5)           Line 322. Didn't the authors want to separate a new subsection?

6)           In Fig. 4, e and f, Did the authors actually calculate statistical significance with n=1 ?

7)           Line 379, 5%CO2 (subscript)

8)           It would help to refer to the Figures in the Discussion to understand the interpretation of the data.

Author Response

In this manuscript, Lowin et al. study the effects of the non-psychotropic cannabinoid CBG on rheumatoid synovial fibroblasts and peripheral blood mononuclear cells. They explore the extent to which CBG mediates the anti-inflammatory effects and how it affects cell viability and antibody production. Moreover, the authors explore how TRPA1 is involved in these processes.

This work was performed with a solid technical competence and the questions addressed by these experiments are clear and interesting. On the other hand, the work suffers from a cluttered graphical presentation of a large number of interesting results. I would recommend reworking the graphic design of the figures so that the meaning of the results is apparent at a glance and the use of colour is strictly purposeful. In addition, because the study contains a number of results whose significance is not entirely clear from the results section, I recommend revising the discussion.

We thank the reviewer for the comments and suggestions. In the discussion we tried to interpret all results including those that were not entirely clear. As the reviewer pointed out, some results are not clear cut but are somewhat murky; therefore, we speculated in the discussion about possible interpretations. We believe, we did this very conservatively and we think that “unclear” results also need to be discussed appropriately.

1)           The exact sample size (n, not just the span) for each experimental group should be given. This applies to all data presented in Figures 1-5. Wherever the mean and standard error are given in the text, n must also be given.

Exact numbers are now given in the figure legends for each conc. and treatment.

2)           Overall, I especially don't like the graphic design of the Figures:

We removed color from Fig. 3 and redesigned Fig. 1 and 5. Data regarding Ca/PoPo under PBS conditions and IgG data from Fig. 5 was moved to supplementary files.

  1. a)           In my copy of the manuscript, the ticks on both axes are lacking in all figures. The correct axis designation must be completed. In addition, I don't understand why there are arrows on the axes in Figs. 1, a to j, and Fig. 2.

Ticks were added, arrows were removed.

  1. b)           In Figures 1, 2, and 3: The time intervals (1038, 3115) that are not equidistant are very unusually chosen. Maybe it would be better to convert sec to min?

We chose three time points due to space restrictions. To keep the distance between time points (except 0) at the same distance, we chose those depicted here. Showing more than 3 time points would make the x axis too crowded and also would not add any more information. We did not interpret data according to certain time points but rather by regarding the whole curve over time. The reviewer is right, however, that sec. is probably not optimal. We changed that to min (rounded).

  1. c)            Moreover, in Figs. 1-3, the labeling of the individual panels with small letters in the middle of the graphs is not clear. I recommend to label the individual panels as it is usually done, on the top left.

We changed that accordingly.

  1. d)           In the legends to all figures, the concentration is given as [c]. This concentration designation is confusing and is not commonly used. At first glance, the legend appears to refer to panel c. I recommend omitting this symbol in the legends and use „concentration“ instead of „[c]“ in the text to figures.

We changed that accordingly.

  1. e)           In the experiments illustrated in Figures 1 and 2, the signals exhibit excessive „noise“ (e.g. panels s,t in Fig. 1). I suspect that the records were exported to some graphics program without rounding the edges of the connecting lines in the software environment, which artificially adds unnecessary variance.

This noise are calcium fluctuations. Under conditions with PBS, where the fluctuations are at maximum, we only investigated 2 patients (more was not necessary due to a lack of effect on TRPA1), rounding edges (looked not much different) still maintained the variation. This variation, however, also changed dependent on stimulus (e.g. TRPA1 antagonist in Fig. 1b vs Fig. 1a).

  1. f)            Figure 3, Comparison of the viability of TNF-exposed and non-exposed cells would be more clear if the y-axes scales were the same in panels a and b. „FBS“ in y-axis description in Figure 3a ? I'm wondering if it might be better to show the y-axis logarithmically in Fig. 3c,d? Why are columns a,b not outlined while c,d are? Are

As for a and b: The starting point (100%) is different for a and b. In a), the 100% line is the value obtained with 0% FBS, no CBG and no TNF. In b), the 100% is the same but with TNF. So, they can’t be really compared even when the scaling is adjusted. Cells in a) started at a lower level then cells in b), as the TNF pre-incubation already boosted proliferation in b). That is why on a per cent basis the rise in a) seems greater although in reality it isn’t.

Now, all sub-figures in Fig. 3 are outlined. A logarithmic depiction made it worse (see image in pdf file).

  1. g)           The labeling of the preincubated CBG in Figure 2b and g is confusing. The low concentration should be explicitly indicated (by indicating 100 nM) and the sentence in the legend to Fig. 2 k and l should be worded differently (not „…without the addition of CBG“)

 This has now been changed. Also, the conc. of all compounds used for pre-incubation have been added.

3)           Throughout the text, I recommend writing specifically TNF-alpha instead of TNF or at least introducing the abbreviation in the abstract/introduction.

TNF (without the alpha) is the official nomenclature as it is the “only” TNF. TNF-beta was renamed to lymphotoxin alpha. The international Universal Protein Resource (UniProt) recommends tumor necrosis factor as the name of the protein (PMID: 27168212, Tumor Necrosis Factor and the Tenacious α).

4)           Line 27, the abbreviation PBMC must be introduced here

This has been added.

5)           Line 322. Didn't the authors want to separate a new subsection?

We now divided this and the subsection is called “TRPA1 influences antibody production”

6)           In Fig. 4, e and f, Did the authors actually calculate statistical significance with n=1 ?

No, we did not include this in our analysis. Initial n numbers were much higher, but TNF could only be detected when PBMCs were stimulated with anti CD3/CD28. We only calculated significance levels for TNF with anti CD3/CD28 and CpG, as the other stimulations often yielded undetectable TNF levels.

7)           Line 379, 5%CO2 (subscript)

Done

8)           It would help to refer to the Figures in the Discussion to understand the interpretation of the data.

This has been added.

Reviewer 2 Report

The authors have studied anti-inflammatory effects of non-psychotropic cannabigerol (CBG) on synovial fibroblasts from rheumatoid arthritis patients in vitro. Manuscript is potentially interesting. My main concerns are:

The observed anti-inflammatory effects were seen at micromolar concentrations, which are unlikely to be achieved after peroral dosing of CBG. As the author correctly discuss, CBG is bound to plasma proteins and only unbound free CBG concentration is pharmacologically active in vivo. Further, CBG has been reported to activate adrenergic a2-receptors with an EC50 value of 0.2 nM i.e. at 4 orders of magnitude lower concentrations than the effects reported here. Intra-articular or topical dosing can possibly reach locally micromolar CBG concentrations. Please discuss/comment.

I am not entirely convinced that CBG induces desensitization of TRPA1-mediated intracellular calcium responses (Figs 2A-E). In our hands several structurally different TRPA1 agonists induces apparent desensitization but this is always due to elevated intracellular calcium baseline. Here baselines are normalized to 100. Are baselines really at similar level even after 30 min pre-incubation with CBG? Please provide absolute values for comparison and revise discussion if needed.

CGB reduced viability strongly without FBS in rheumatoid synovial fibroblasts. Thus the reduction seen on IL-6 and IL-8 cytokines in experiments having low FBS concentration might be due to low cell numbers/viability issues. Please discuss/clarify.

Sentence on lines 153-155: “Without FBS, TNF-induced IL-6 and IL-8 production was almost completely abrogated by CGB (3.125-50 µM) but concentration dependent effects were not observed (Fig. 3c,d).” could be removed or rephrased.

Sentence on line 157: “CBG inhibited IL-8 only at 50 µM (Fig. 3d)” could be rephrased

Author Response

The authors have studied anti-inflammatory effects of non-psychotropic cannabigerol (CBG) on synovial fibroblasts from rheumatoid arthritis patients in vitro. Manuscript is potentially interesting. My main concerns are:

The observed anti-inflammatory effects were seen at micromolar concentrations, which are unlikely to be achieved after peroral dosing of CBG. As the author correctly discuss, CBG is bound to plasma proteins and only unbound free CBG concentration is pharmacologically active in vivo. Further, CBG has been reported to activate adrenergic a2-receptors with an EC50 value of 0.2 nM i.e. at 4 orders of magnitude lower concentrations than the effects reported here. Intra-articular or topical dosing can possibly reach locally micromolar CBG concentrations. Please discuss/comment.

The reviewer is right; we also thought that concentrations higher than 1-5µM cannot be achieved in vivo by oral or i.p. application of CBG. However, this study (PMID: 21796370) assessed in vivo concentrations in mice and rats after application of CBG or CBD. They found plasma levels after incubation with a very high dose of CBG (120mg/kg) to be 120µM [40.8µg/mL] with i.p. administration and around 2µM [0.67µg/mL] with oral dosing. In rats, this study found somewhat lower levels, but they were still in the µM range. The half-life was around 3h. In addition, repeated dosing might also boost in vivo levels, as CBD also has low bioavailability, but long-term treatment might lead to accumulation of the compound in certain tissues. As the reviewer pointed out, intraarticular injections might also be a viable option to selectively increase CBG conc. in synovial tissue.
As for the effects at α2-ARs: It is true that CBG has been described as an agonist at this receptor but when closely looking at available in vivo data, activation of α2-AR seems questionable. The use of α2-AR agonists is usually accompanied by severe sedation and analgesia but CBG even at 100mg/kg did not show any sedative effects. This study (PMID: 35899583) investigated pain responses and found that 10mg/kg of CBG attenuated pain in an α2-dependent manner but the main effect was mediated via CB2.  The results obtained with the α2 ligand used in this study (atipamezole) were not properly interpreted as it is an inverse agonist, that can also increase pain thresholds just by itself. In another study, food intake after CBG administration was assessed (PMID: 27503475). Even though they used very high [c] of CBG (up to 240mg/kg) they did not detect any motoric side effects or sedative effects. In fact, CBG had a stimulant effect on locomotor activity. All these above mentioned effects would occur with a prototypical α2-AR agonist. In addition, there is no study that actually shows a decrease in norepinephrine release in response to CBG. Lastly, the original study (PMID: 20002104) that identified CBG as an α2-AR agonist also shows that CBG is a very low efficacy agonist at α2-AR. When comparing binding of [35S]GTPγS after CBG stimulation, it shows that its effect at cannabinoid receptors is much stronger than at α2-ARs.
By the way, CB2 involvement in the effect of CBG can be ruled out, as we already tested several CB2 ligands and found no influence on cytokine production. In fact, CB2 on synovial fibroblasts elicits pro-inflammatory effects (PMID: 30943136) and this study also showed no effect of CB2 agonists on cytokine levels.

I am not entirely convinced that CBG induces desensitization of TRPA1-mediated intracellular calcium responses (Figs 2A-E). In our hands several structurally different TRPA1 agonists induces apparent desensitization but this is always due to elevated intracellular calcium baseline. Here baselines are normalized to 100. Are baselines really at similar level even after 30 min pre-incubation with CBG? Please provide absolute values for comparison and revise discussion if needed.

We were also surprised about this finding and therefore we showed absolute values in Fig. 2K. Here we demonstrate that a 30min incubation with 100nM CBG does not influence basal calcium levels. However, we cannot rule out that Ca2+ mobilization occurs in microdomains that might elicit this desensitizing effect. Also, it could well be that even if we only incubated with 100nM CBG, the intracellular levels are much higher thus allowing desensitization. We also added this to the discussion. 

CGB reduced viability strongly without FBS in rheumatoid synovial fibroblasts. Thus the reduction seen on IL-6 and IL-8 cytokines in experiments having low FBS concentration might be due to low cell numbers/viability issues. Please discuss/clarify.

 This is exactly what we found and also discussed in the manuscript (see lines 274-277). At low or no FBS conditions, CBG was much more efficacious regarding suppression of cytokine production and this was associated with reduced cell viability; however, at 10% FBS, the reduction of cytokine levels was independent from cell viability. In Fig. 2A and B we show that at 10% FBS only the highest [c] of CBG is able to reduce cell viability.

Sentence on lines 153-155: “Without FBS, TNF-induced IL-6 and IL-8 production was almost completely abrogated by CGB (3.125-50 µM) but concentration dependent effects were not observed (Fig. 3c,d).” could be removed or rephrased.

Done

Sentence on line 157: “CBG inhibited IL-8 only at 50 µM (Fig. 3d)” could be rephrased

Done

Round 2

Reviewer 1 Report

The authors have responded to all my comments. Thank you.

Reviewer 2 Report

I have following comments to author responses.

Even if CBG total plasma concentration is 2 uM most of it is highly bound to plasma proteins. I could not find data for CBG plasma protein binding (actually this should be the authors task), but data is available for cannabidiol (CBD), a related cannabinoid, that is more than 88% bound to plasma proteins. If CBG binds similar to CBD to plasma proteins, then free pharmacologically active unbound concentration of CBG is 240 nM (i.e. 12% free concentration of the total).

A recent study described biased a2 adrenergic agonists that are not sedative although such compounds penetrate well into the brain (PMID: 36173843). It would be interesting to know in future if CBG is biased adrenergic a2 agonist too. Published data shows that CBG shows variable penetration into the brain depending on administration route. Limited BBB penetration may partly explain lack of sedation through adrenergic a2 agonism by CBG. Atipamezole is a highly potent non-subtype selective adrenergic a2 antagonist, not inverse agonist, developed Orion Pharma/Farmos where the reviewer is currently employed and is highly knowledgeable about pharmacology of atipamezole.

In future publication the authors are advised to take close look on the pharmacologically active metabolites of CBG, that may contribute to net pharmacological effects evoked by CBG in vivo. Unfortunately, the authors did not bother to discuss or study biologically active metabolites of CBG. Perhaps this could be something for the next study.

https://chemrxiv.org/engage/api-gateway/chemrxiv/assets/orp/resource/item/62c5fd7e14201f3bbe287333/original/metabolites-of-cannabigerol-cbg-generated-by-human-cytochrome-p450s-are-bioactive.pdf

Despite these failings, I recommend acceptance of the present version of the MS.